# Malaria Severity in the Elimination Continuum: A Retrospective Cohort Study between Beitbridge and Lupane Districts in Zimbabwe, 2021–2023

**DOI:** 10.3390/ijerph21070877

**Published:** 2024-07-04

**Authors:** Same Betera, Bambang Wispriyono, Wilfred Njabulo Nunu, Dewi Susanna, Nicholas Midzi, Patience Dhliwayo, Fitra Yelda, Melisa Nyamukondiwa

**Affiliations:** 1Department of Environmental Health, Faculty of Public Health, Kampus Baru Depok, Universitas Indonesia, Kota Depok 16424, Jawa Barat, Indonesia; sambetera@gmail.com (S.B.); dsusanna@ui.ac.id (D.S.); nyamukondiwamel@gmail.com (M.N.); 2Environmental Health Department, Ministry of Health and Child Care, Kaguvi Building, 4th Floor, Causeway, Harare P.O. Box CY 1122, Zimbabwe; 3Department of Environmental Health, Faculty of Health Sciences, School of Public Health, University of Botswana, Gaborone UB 0022, Botswana; njabulow@gmail.com; 4Department of Public Health, Faculty of Health Sciences, University of the Free State, Bloemfontein P.O. Box 339, South Africa; 5National Institute of Health Research (NIHR), Ministry of Health and Child Care, 65 Josiah Tongogara, Harare Street, Harare P.O. Box CY 1122, Zimbabwe; 6National Malaria and Control Program, Ministry of Health and Child Care, Kaguvi Building, 4th Floor, Causeway, Harare P.O. Box CY 1122, Zimbabwe; 7Research Center of Health Science, Faculty of Public Health, Kampus Baru Depok, Universitas Indonesia, Kota Depok 16424, Jawa Barat, Indonesia

**Keywords:** malaria elimination, malaria severity, resurgence, travel history, vulnerability

## Abstract

Malaria has created a resurgence crisis in Zimbabwe’s elimination continuum, diverging from global commitment to malaria elimination by 2030. This retrospective cohort study aimed to determine the risk factors associated with severe malaria in the Beitbridge and Lupane districts. Multistage sampling was used to recruit 2414 individuals recorded in the District Health Information Software2 Tracker database. The study used IBM SPSS 29.0.2.0(20) for data analysis, and odds ratios (ORs) to estimate the relative risk (RR; 95% C.I; *p* < 0.05). The study revealed significant relative risks (*p*-value < 0.05) for individuals who had no Long-Lasting Insecticidal Nets (Beitbridge 47.4; Lupane 12.3), those who owned but used the LLINs (Beitbridge 24.9; Lupane 7.83), those who slept outdoors during the night (Beitbridge 84.4; Lupane 1.93), and adults (Beitbridge 0.18; Lupane 0.22) compared to the corresponding reference groups. Other factors showed varying RR: sex (Beitbridge 126.1), prompt treatment (Beitbridge 6.78), hosting visitor(s) (Lupane 6.19), and residence (Lupane 1.94) compared to the corresponding reference groups. Risk factor management needs to focus on increasing local awareness of malaria, universal LLINs coverage of indoor and outdoor sleeping spaces, community-based programs on proper and consistent LLIN usage, screening of visitors from malaria-endemic areas, comprehensive entomological activities, mixed malaria interventions in rural hotspots, and future research on local malaria transmission dynamics. While Zimbabwe has the potential to meet the global goal of malaria elimination, success depends on overcoming the risk factors to sustain the gains already made among malaria elimination districts.

## 1. Introduction

The resurgence of malaria has remained high in many malaria-free areas [1,2]. The World Health Organization (WHO) defines reintroduction as the occurrence of an outbreak or reestablishment of endemic malaria in previously eliminated areas [2]. Malaria can be classified as imported, introduced, or indigenous in elimination settings [3]. The WHO further states that malaria elimination areas must demonstrate that no new indigenous cases have occurred in the last three years [3].

Malaria is the most prevalent parasitic infection caused by five predominant species of *Plasmodium* (*P. falciparum*, *Plasmodium vivax*, *P. ovale*, *P. malariae*, and *P. knowlesi*) that have been documented in eighty-seven countries to cause substantial morbidity and mortality in humans globally [4,5,6]. In 2021, the global malaria landscape reported approximately 247 million cases, with a slight increase from the 2020 figure of 245 million cases [7]. Africa accounts for approximately 95% of global malaria cases, with an estimated 234 million cases by 2021 [7]. Notably, the COVID-19 pandemic disruptions between 2019 and 2021 contributed to an additional 13.4 million malaria cases [5,6]. The global malarial mortality rate has undergone significant changes over the years [7].

The WHO states that unless there is a system of vigilance with a properly formed epidemiological service and an efficient organisation for timely remedial actions, there is a likelihood that malaria will re-establish itself in regions where the disease has been eliminated [8,9]. Vigilance is a critical component of malaria surveillance after elimination [10] and must be developed before malaria elimination endorsement can be issued [11]. Achieving global and national goals for malaria eradication requires the adaptation of effective elimination strategies to ensure continued progress [8] and these strategies must be implemented swiftly to reduce malaria morbidity and mortality [12]. However, these interventions encounter increasing obstacles that contribute to malaria [13,14]. Governments have recently prioritised the prevention of non-indigenous malaria transmission as part of their programs to reduce the occurrence of the disease [15].

Malaria remains a major public health problem, with more than 5 million people [16] in Zimbabwe at risk of contracting it annually [16]. In Zimbabwe, most malaria cases and deaths occur in moderate-to high-transmission districts. Malaria reduction has been very rapid in the southern and central parts of the country where the malaria incidence has declined to 5 cases per 1000 persons and below; however, the risk of malaria in Zimbabwe has increased towards the country’s borders with Mozambique and Zambia [17,18,19]. 

In addition to its health implications, malaria has a significant socioeconomic impact on Zimbabwe [13]. During the strategic period of 2016–2020, the years 2017 and 2020 exhibited the highest burden of malaria, with national incidence rates of 34 and 32 cases per 1000 individuals per year, respectively [20]. In 2022, the country recorded 140,753 confirmed cases of malaria, resulting in an incidence rate of 9 cases per 1000 person [20]. While the 2022 incidence remained the same as that in 2021, malaria deaths increased significantly by 36% from 131 to 177, resulting in a mortality rate of 1.2 per 100,000 persons [20].

Over the past 13 years, the Zimbabwean Ministry of Health and Child Care (MoHCC), through the National Malaria Control Program (NMCP), in tandem with the Roll Back Malaria (RBM) Initiative, has made remarkable strides in addressing malaria challenges [16]. Together, they forged a path towards progress, demonstrating a shared vision for malaria-free Zimbabwe [21,22]. This ambition underscores the nation’s determination to safeguard the health and well-being of its citizens, reduce the economic burden imposed by malaria, and ensure a brighter and healthier future for generations [19].

In line with its vision of becoming a malaria-free country, Zimbabwe began implementing malaria pre-elimination activities in 2012, starting with seven districts (11%) in Matabeleland South, including Beitbridge [19]. By 2015, Zimbabwe had transitioned to an additional 13 low-burden districts in Matabeleland North, Bulawayo, the Midlands, and Mashonaland West Provinces to pre-elimination status, including Lupane as 1 of 5 elimination districts in Matabeleland North [18]. By the end of 2022, the country had increased the number of pre-elimination districts to 30 (48%) of 62 [18].

Zimbabwe’s malaria elimination focus investigation and response guidelines, which were rolled out in 2017, provide guidance and standardise data analysis, focus delimitation, classification, determination of drivers of transmission, reporting, and response activities aimed at addressing the drivers of transmission. The country adopted a 1-3-7 surveillance model whereby this approach targets reporting of confirmed cases within one day, investigation and classification of specific cases within three days, and targeted control measures to prevent further transmission within seven days [23].

Beitbridge and Lupane persistently experience the highest number of malaria cases among the World Health Organisation-certified malaria elimination areas in Zimbabwe. The cumulative data for both districts reveal an undesired trend, with the total number of malaria cases increasing by approximately 319% from 959 in 2021 to 4023 in 2023 [20]. Between 2021 and 2023, the Beitbridge and Lupane districts reported 5746 malaria cases, of which 5127 were local cases and 619 were imported cases [20]. In the Beitbridge district, a previous study suggested a significant assumed risk for locally acquired malaria, with an odds ratio (OR) of 2.7 compared to imported cases. The study further revealed that out of 75 cases, 63 (84.0%) had no travel history exposure at an average of six weeks prior to malaria diagnosis [21].

In line with this background, this retrospective cohort study took a systematic approach to determine the risk factors associated with malaria severity in two selected elimination districts, Beitbridge (Matabeleland South Province) and Lupane (Matabeleland North Province) in Zimbabwe for the period covering 1 January 2021 to 31 December 2023. The specific objectives of this study were as follows:Contrast variations in socio-demographic characteristics that affect malaria severity;Compare the role of malaria prevention practices on malaria severity;Assess the association between travel history and malaria severity using multivariate logistic regression models.

### Operational Definitions

Prompt treatment: Appropriate and prompt malaria treatment was administered within 24 h of fever onset.

Severe malaria: Severe malaria is defined as the presence of *Plasmodium falciparum* parasitaemia and is characterised by the presence of one or more clinical or laboratory features, indicating a risk of complications or organ dysfunction. Major complications of severe malaria include cerebral malaria, pulmonary oedema, acute renal failure, severe anaemia, and/or bleeding. Acidosis and hypoglycaemia are the most common metabolic complications. 

Uncomplicated malaria: This refers to cases of malaria with no clinical or laboratory signs of severe disease. Symptoms typically include fever, chills, headaches, muscle aches, and fatigue.

Vulnerability: The frequency of influx of infected individuals or groups and/or infective anopheline mosquitoes

## 2. Materials and Methods

### 2.1. Study Design and Sampling

This retrospective cohort study determined the risk factors associated with severe malaria. The study adopted the measurement of travel history exposure as reported in by the secondary data using (i) within-country travel to malarious areas, (ii) foreign travel (travel to places beyond the country’s borders), and (iii) travel within six weeks of malaria before diagnosis. 

(i)Exposed group: Individuals in this group were traced to have contracted malaria from a known malarious area outside the elimination district including areas beyond the country’s borders. The DHIS2 tracker electronic database names the data element “Malaria cases imported”.(ii)Unexposed: Individuals in this group were traced to have contracted malaria in the reporting district. The DHIS2 tracker electronic database names the data element “Malaria cases local”.

The dependent variable was malaria severity, with binary outcomes for uncomplicated (0) and severe (1) malaria cases. The independent variables were grouped into sociodemographic risk factors (district, age, sex, residence, host visitor(s), and occupation) and malaria prevention practices (prompt treatment, malaria contact status, malaria species infection, LLIN use, and night outdoor sleeping behaviour). This study transformed all variables into dummy variables, enabling the inclusion of categorical predictors in binary logistic regression and the interpretation of their impact on the outcome variable. Children under five are more likely to experience severe symptoms such as cerebral malaria, severe anaemia, and respiratory distress. These clinical differences justify the need for separate categorisations to effectively tailor interventions and treatment protocols. However, the present study assigned codes (0) to individuals <5 years and codes (1) to individuals ≥5 years, since this category included school-aged children and adults who had different patterns of exposure to the risk factors, compared to the under-5-year-old group, and regardless of the perceived information, that, the under-5 group is always at a greater risk. Similarly, for the sex variable, the study assigned code (0) to females and code (1) to males. Before receiving antimalarial treatment, rapid diagnostic tests (RDTs) of a blood sample or a microscopic examination of blood films confirmed the malaria diagnosis in all individuals in this study. In this study, the malaria parasite species were categorised as follows: *Plasmodium falciparum* was coded (2), *Plasmodium malariae* was coded (1), and both *Plasmodium vivax* and *Plasmodium ovale* were combined into one category, “other species”, and coded (0). Notably, the secondary data used in the study recorded no individuals infected with *Plasmodium knowlesi* (Appendix A: Details on other variables).

The study used multistage stratified random sampling to recruit 2414 (42%) all-sex and age-group individuals comprising 2112 local malaria cases and 302 imported malaria cases from a total of 5746 individuals. This enrolment covered the period from 1 January 2021 to 31 December 2023 in two selected malaria elimination regions, the Beitbridge and Lupane districts. The current study employed the Fleiss formula with a continuity correction (CC) factor using O.R = 2.7, assuming that 84.0% of the individuals in the unexposed group suffered from malaria [21]. Using secondary data on imported and local cases, the study found an assumed ratio (1:7) for the exposed and unexposed DHI2 tracker databases. The present study used a 95% confidence level and a 90% power level. Subsequently, the parameters were combined to compute the sample sizes in Epi InfoTM for Windows version 7.2. This calculation yielded a total sample size for each district of 1056 local malaria cases and 151 imported malaria cases. To ensure adequate representation of each subgroup in the sample for each district and year (2021, 2022, and 2023), proportional stratified random sampling was employed. The sample size was determined to ensure a significant representation of the exposed and unexposed samples as well as a statistical proportion for comparison during model building. This approach provided a robust framework for comparing risk factors associated with severe malaria between the two districts.

### 2.2. Study Sites

Zimbabwe, a southern African country with ten provinces, shares borders with five neighbouring countries. The Zimbabwe National Malaria Control and Roll Back Malaria Programmes had successfully eliminated malaria in 30 (48%) districts by the end of 2022 [17,24]. Matabeleland South Province, which is located in the southern part of the country, includes Beitbridge as one of its five districts in the agro-ecological region [21,25,26,27,28]. The Lupane district is the provincial capital of Matabeleland North Lupane, characterised as semi-arid in agro-ecological region four [28,29]. The rural wards of Lupane share borders with the Binga district, where malaria transmission is predominantly unstable and mesoendemic [30]. Economic activities in both Beitbridge and Lupane districts include formal employment, tourism, large-scale farming, small-scale farming, artisanal mining, fishing, cross-border trading, vending, drought-resistant cropping, and livestock husbandry. Vulnerable groups often operate in temporary structures, which makes them unsuitable for indoor residual spraying. The dynamic nature and mobility of the population complicate the distribution of Long-Lasting Insecticidal Nets (LLINs), necessitating repeated distribution exercises [21,24,25,26]. A map of the study area is shown in Figure 1.

### 2.3. Ethics and Data Collection

The study received ethical approval from the Universitas Indonesia (Ket-26/UN2.F10.D11/PPM.00.02/2024), the Zimbabwe Medical Research Council (MRCZ/B/2629), and the Ministry of Health and Child Care. As this study utilised secondary data, obtaining informed consent from individuals was not feasible. This study ensured privacy and confidentiality through data anonymisation. The main data source for this study was the District Health Information Software 2 tracker (DHIS2 tracker) database for Zimbabwe. Routine and non-routine sources with historical data on patients, such as outpatient department (OPD) registers, medical charts, tally sheets, malaria registers, laboratory registers, travel history records, occupation records, program reports, survey reports, and other relevant documentation, were used to supplement the main data sources. Following the export of relevant data to the master Excel database, a systematic process of collation and analysis was undertaken to ensure the completeness, correctness, and consistency of the data.

### 2.4. Validity and Reliability

To complement the DHIS2 tracker database and enhance the validity and reliability of measurements, this study employed standardised variables by leveraging the 2018 WHO Malaria Surveillance, Monitoring, and Evaluation Reference Manual, as well as Zimbabwe’s 2017 Foci Investigation and Response Guidelines for Malaria Elimination.

### 2.5. Statistical Analysis

#### 2.5.1. Univariate Analysis

The collected quantitative data were analysed using the Statistical Package for the Social Sciences [IBM SPSS 29.0.2.0(20)] for Windows. The researchers double-checked and cleaned the imported data. The study used descriptive statistics to present an overview and summary of the key variations in malaria risk factors among the study groups. This study used both the mean and median as key measures of central tendency to summarise the data on age and prompt treatment variables.

#### 2.5.2. Bivariate Analysis

A bivariate analysis was used to compare the relationships and identify each strong independent variable associated with severe malaria. The study first used cross-tabulations and subsequently the chi-square method to test for a significant association (*p*-value < 0.05) between each independent variable and the outcome variable. The study applied the Fisher exact method when there was a violation of the chi-square assumption, specifically, when the number of cells with expected frequencies below five exceeded 20% of the total cells. A bivariate logistic regression was used to examine the extent to which each independent factor contributed to variation in the outcome variable. 

#### 2.5.3. Multivariate Binary Logistic Regression Analysis

Since the prevalence of malaria is less than 10% within the elimination districts in Zimbabwe [21], the study employed odds ratios (ORs) to approximate the relative risks (RRs) and assess the association between each risk factor and malaria severity.

(i)Bivariate selection: The chi-squared test was employed from the omnibus test of the model coefficients to assess the significance of each independent variable. If an independent variable was obtained (*p*-value ≤ 0.25), the present study considered that the variable contributed significantly (effect) to the model in explaining variability in malaria severity and proceeded to the multivariate modelling. However, the study considered important variables with a *p*-value >0.25 for the multivariate logistic analysis based on the literature.(ii)Backward stepwise selection (full model): Our study used backward stepwise selection using the primary independent variable (travel history exposure with no (0) and yes (1)), the dependent variable (malaria severity with binary outcomes for uncomplicated malaria (0) and severe malaria (1)), and all the confounder variables regardless of their significance. The study considered a variable to be a confounder and returned it to the model if its removal caused a change in the estimated RR value of the remaining variables in the full model of more than ten percent (>10%). As the prevalence of malaria is less than 10% within the elimination districts in Zimbabwe [21], the multivariate regression analyses presented results in the form of odds ratios (ORs) along with their corresponding 95% confidence intervals (CIs), and statistical significance was determined by a *p*-value < 0.05.(iii)Interaction tests among independent variables were conducted by multiplying the values of the two independent variables involved in the interaction (*x*_1_*.x*_2_) and assessing whether the effect of one predictor variable on the outcome variable depended on the level of another predictor variable. The Wald test was used to determine the significance of the interaction term (*p* < 0.05, implying an interaction effect).(iv)Model evaluation: Our study utilised the omnibus, pseudo-parameters of the Nagelkerke R-squared, and Hosmer–Lemeshow tests for the model evaluation. The omnibus test assessed the overall fit of the logistic regression model by testing the null hypothesis that all regression coefficients were equal to zero. A significant omnibus test (*p* < 0.05) indicated a good overall fit for the data. Nagelkerke R-squared pseudo-parameters quantified the variation explained by the model, with values closer to one (1) indicating a stronger relationship between the predictors and malaria severity. The Hosmer–Lemeshow test assessed the model’s goodness of fit, with a non-significant result (*p*-value > 0.05) suggesting good calibration and fit to the data. These tests collectively helped us assess the adequacy, explanatory power, and predictive accuracy of the logistic regression model for malaria severity, based on the given predictors.

## 3. Results

The overall results showed that the majority of individuals (n = 1465, 61%) were male. The predominant age group was five years and above, with similar trends observed in Beitbridge and Lupane, with 1130 (94%) and 1085 (90%), respectively. The overall distribution based on residence location showed that the majority lived in rural areas, accounting for 1602 (66%) of the study population. *Plasmodium falciparum*, comprising 2096 cases, stood as the overall predominant cause of infection, represented 87% of all cases and was the predominant cause of infection (Appendix A). 

In the Beitbridge district, the cross-tabulation results revealed that males were 95.1 [(95% CI: 23.4–386); *p* < 0.001] times more likely to develop severe malaria than females. Individuals who sought treatment after 24 h were 7.32 [(95% CI: 4.83–11.1), *p* < 0.001] times more likely to develop severe malaria than those who sought treatment within 24 h. The results showed that individuals who had no LLINs were 5.79 [(95% C.I: 3.11–10.8), *p* < 0.001] times more likely to develop severe malaria than those who had LLINs and used them (Appendix A). 

In the Lupane district, the cross-tabulation results revealed that individuals who had visitors were 5.64 [(95% CI: 4.00–7.96]; *p* < 0.001) times more likely to develop severe malaria than those who did not. The relative risk (RR) of severe malaria among individuals infected with *Plasmodium falciparum* was 8.56 [(95% CI: 1.16–63.2]; *p* = 0.035) times higher than that of the reference group in the Lupane district. In that district, individuals who had no LLINs were 10.7 [(95%C.I: 6.35–17.9), *p* < 0.001] times more likely to develop severe malaria than the individuals who had LLINs and used them, respectively (Appendix A).

Bivariate selection results revealed that nine (9) variables were significantly associated with the outcome at a *p*-value of less than 0.25. The variables residence and hosting visitor(s)’ obtained *p*-value was more than 0.25 at the bivariate selection but these variables were considered important to continue into multivariate modelling for the Beitbridge district. In the Lupane district, ten (10) variables were significantly associated with the outcome at a *p*-value of less than 0.25. The variable prompt treatment had a value higher than 0.25 but was considered important to continue the multivariate analysis (Appendix A). 

In the overall multivariate analysis, ten (10) variables, namely, district, age group, gender, occupation, visitor(s), residence, prompt treatment within 24 h, malaria species, LLIN use, and sleeping outdoors during the night, were the confounding variables (>10% percentage change) of the relationship between travel history and malaria severity. Individuals with a travel history were approximately RR ≈ 0.18 [(95% CI: 0.05–0.70, *p* = 0.013)] times less likely to develop severe malaria than those without a travel history exposure after controlling for the above confounding variables (Appendix A). The study further obtained results from a stratification analysis by district (Table 1).

The overall results across Models I, II, and III indicated strong model fits as shown in Table 2. In Model I, the omnibus test of model coefficients (χ^2^ = 733.6, *p* < 0.05), supported by the Hosmer–Lemeshow test (*p* = 0.121), demonstrated a good fit, with an overall classification accuracy of 87.8%. Model II, which focused on the Beitbridge district, also showed a strong fit with 95.9% accuracy and a substantial variation explained (36–76%). Similarly, Model III, targeting the Lupane district, demonstrated a good fit (χ^2^ = 412.1, *p* < 0.05) with 85.0% accuracy and an explained variation ranging from 29% to 45% (Cox and Snell R-squared = 0.289, Nagelkerke R-squared = 0.452).

## 4. Discussion

This study was the first to investigate the association between travel history and malaria severity in WHO-certified malaria-free zones in Zimbabwe. To the best of our knowledge, this study is also the first to compare and analyse data from two malaria elimination districts within two different provinces, Beitbridge (Matabeleland South Province) and Lupane (Matabeleland North Province), which initiated malaria elimination in the country in 2012 [19].

### 4.1. Demographic Characteristics

The study revealed that age group was a significant confounder of the relationship between travel history and malaria severity. A similar trend was that the risk of developing severe malaria was lower among adults than among those under 5 years of age in Beitbridge and Lupane. These findings indicate an interplay between travel history and age-related exposure to malaria. Children under five travel less frequently than adults, which aligns with the finding that individuals without a travel history are more likely to develop severe malaria compared to those with a travel history. Children under five are more susceptible to malaria because their immune systems are still developing. In addition, they are often less likely to use preventive measures effectively due to their dependency on caregivers for their implementation, such as sleeping under treated mosquito nets, which emerged as a dominant risk factor in the present study. However, these findings contradict a study conducted in Batu town (Ethiopia), which found that the highest malaria infection rates were among individuals older than 14 years, and this difference was statistically significant [31]. 

Males were more likely to develop severe malaria than females after accounting for confounding variables in the Beitbridge district, whereas there was no significant association between gender and severe malaria in the Lupane district. The Beitbridge district is an important trade route connecting Zimbabwe to several countries [32]. As a result, cross-border livelihood activities might have contributed to the higher risk of severe malaria among males than among females. In the same district, the significant results on late treatment-seeking and low use of treated mosquito nets suggest that these factors could have resulted in synergistic effects, possibly increasing the likelihood of severe malaria among males. Similar to the Lupane district, a previous study conducted in China highlighted that the difference in gender distribution was not statistically significant, with a χ^2^ value of 3.531 and a *p*-value of 0.473 [33]. These contrasting results across the two districts emphasise the potential influence of location-specific and livelihood factors on the risk of severe malaria in males and females.

The current study found that individuals residing in rural areas were more likely to develop severe malaria than those in urban areas after accounting for confounding variables in Lupane, whereas the residence distribution was not statistically significant in Beitbridge. Several factors might have contributed to the prevalence of severe malaria in rural Lupane. The rural wards of Lupane share borders with the Binga district, where malaria transmission is predominantly unstable and mesoendemic [30]. As a result, malaria transmission may be high in border areas, leading to more severe malaria cases in Lupane’s rural wards than in its urban wards. Major rivers, such as the Gwayi and Shangani rivers, intersect in Lupane Rural, flowing northward and meeting at the Gwayi Shangani Dam [29]. These bodies of water create an abundance of mosquito breeding sites, which significantly contribute to the local transmission of malaria. Consequently, people living near these water bodies face a greater risk of contracting malaria from infected mosquitoes, leading to a higher prevalence of severe malaria cases in these rural areas. The findings in Beitbridge concur with a previous study in Rwanda, which found that the percentage of severe malaria cases in rural areas (18.9%) was only slightly lower than that in urban areas (20%), with no statistically significant difference observed [34].

There was no significant association between visitors and malaria severity in Beitbridge. However, individuals who had visitors were more likely to develop severe malaria than those who had no visitors in Lupane. This difference suggests that visitors from areas with a higher prevalence and already infected with malaria might have increased the transmission of the disease to local populations in Lupane. Similarly, a study in China found that among malaria cases, approximately 50.3% were migrant workers and an additional 19.1% were visitors who had come to see their relatives or friends, with the difference being statistically significant (Fisher’s Exact Test, *p* < 0.001) [33]. However, future researchers should consider investigating the contribution of malaria transmission by visitors, especially in elimination settings.

### 4.2. Malaria Prevention Practices

The study revealed a shorter median time to seek malaria treatment in Beitbridge (1.0 ± 2.2 days) compared to Lupane (2.6 ± 2.4 days), indicating potential differences in healthcare access or awareness between the districts. In contrast, a study in Henan Province, China, reported a median duration of three days between initial symptoms and diagnosis, highlighting a longer variation in healthcare systems and diagnostic processes [12]. Individuals who received malaria treatment 24 h after symptom onset in Beitbridge were more likely to progress to severe malaria than those who sought treatment within 24 h after controlling for confounders. The difference in prompt treatment distribution was not statistically significant in the Lupane group. The exceptionally high relative risk in Beitbridge compared to Lupane suggests that delayed treatment posed a much greater risk in Beitbridge, and the significant differences observed could stem from various variables, such as healthcare systems and malaria education in the population.

The current study revealed that individuals who had no LLINs, as well as individuals who had LLINs but had not used them, had a higher risk of developing severe malaria than those who had LLINs and had used them after adjusting for confounders in Beitbridge and Lupane. Earlier studies consistently agree that individuals who do not use bed nets face a greater risk of contracting malaria, particularly those treated with long-lasting insecticides (LLINs). By establishing a protective barrier around those sleeping beneath them, LLINs reduce malaria-related illnesses and mortality and provide substantial protection against malaria infections in individuals of all ages [31,35,36]. 

Another finding was the association between sleeping outdoors at night and severe malaria. Our study found that the risk of severe malaria was higher among individuals who slept outdoors during the night than among those in Beitbridge and Lupane, after accounting for confounding variables. These findings are supported by a matched case-control study conducted in the Ziway-Dugda district (Ethiopia), which highlighted that staying outdoors late at night (AOR = 2.99) was among the determinants of malaria [37]. The risk in Beitbridge was marginally higher than in Lupane, and a combination of factors may have accounted for the remarkably high relative risk of severe malaria among individuals who slept outdoors in Beitbridge. These factors could include cross-border activities and indigenous livelihood practices, which could have increased malaria vector exposure and transmission in the region. 

### 4.3. Evidence to Malaria Resurgence due to Local Transmission 

Individuals with a travel history were approximately RR ≈ 0.04 [(95% CI: 0.005–0.379), *p* = 0.004) times less likely to develop severe malaria than those without a travel history exposure after controlling for the above confounding variables in Beitbridge. The logistic regression model yielded a goodness-of-fit Cox and Snell R-squared of $R_{cs}^{2}=$ 0.356 and Nagelkerke R-squared of $R_{N}^{2}=$ 0.745. These values indicated that the set of predictors in this model explained between 36% and 75% of the variation in the dependent variable. Similarly, individuals with a travel history were approximately RR ≈ 0.03 [(95% CI: 0.010–0.114), *p* < 0.001) times less likely to develop severe malaria than those without travel history exposure in Lupane. The logistic regression model yielded a goodness-of-fit Cox and Snell R-squared of $R_{cs}^{2}=$ 0.289 and Nagelkerke R-squared of $R_{N}^{2}=$ 0.452. These values indicate that the set of predictors in this model explained 29–45% of the variation in the dependent variable. 

Building on these findings, the study rejects the null hypothesis at a 95% confidence level and concludes that individuals with a travel history have a lower likelihood of developing severe malaria than those without a travel history, suggesting the resurgence of malaria in the selected elimination districts. Conversely, a meta-analysis of 22 studies in pre-elimination settings in sub-Saharan Africa found that travel exposure was a major risk factor for malaria. The pooled odds ratio (OR) was 3.77 (95% CI: 2.49–5.70) [38]. 

The findings of the present study indicate that local transmission, rather than imported cases, primarily links severe malaria cases and identifies vulnerability factors within the study population that intensify the risk of severe malaria. These findings emphasise the intricate interplay between malaria dynamics, suggesting that the resurgence of malaria in elimination settings can be propelled by local cases under conducive conditions of receptivity and vulnerability. Understanding the dynamics of local transmission is critical for developing public health strategies aimed at eliminating malaria. The persistence of local transmission and its contribution to severe malaria underscores the challenges of achieving and sustaining malaria elimination in the Beitbridge and Lupane districts. 

## 5. Conclusions

In view of these findings, risk factor management needs to focus on increasing local awareness of malaria prevention, universal LLIN coverage of indoor and outdoor sleeping spaces, community-based programs on proper and consistent LLIN usage, screening of visitors from malaria-endemic areas, comprehensive entomological activities, and mixed malaria interventions in hotspots. Continued research on local transmission dynamics, including factors influencing malaria receptivity and vulnerability, is crucial. While Zimbabwe has the potential to meet the global goal of malaria elimination, success depends on overcoming the risk factors to sustainably interrupt local transmission in the elimination provinces. Despite the study’s findings, both local and imported cases of malaria need equal attention to sustain the gains made in malaria elimination.

## 6. Strengths and Limitations

This retrospective cohort study had several strengths and limitations. Data on malaria resurgence using two selected elimination districts from two provinces that initiated malaria elimination efforts in Zimbabwe were analysed. The 3-year study period facilitated a longitudinal analysis and indirectly incorporated seasonal variations in malaria transmission and severity. The study accessed extensive DHIS2 databases, providing a study with a large sample size and enhanced statistical power. Additionally, this study generated hypotheses for future research by identifying potential associations between exposure and outcomes. However, this study used historical data, which varied in quality and completeness. To address this limitation, the study used other medical records to complement the database and employed a statistical analysis to address the missing data.

## Figures and Tables

**Figure 1 ijerph-21-00877-f001:**
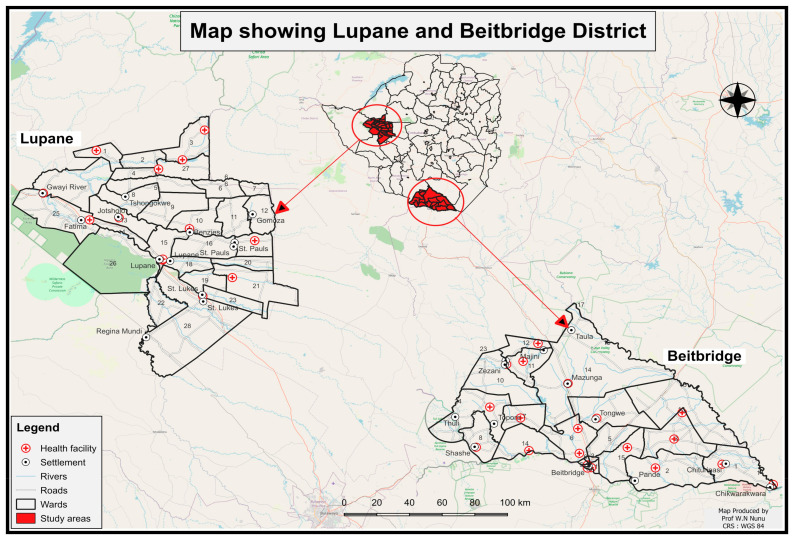
Map of the study area.

**Table 1 ijerph-21-00877-t001:** Malaria severity stratified by district.

Variables	Category	n (%)	Beitbridge District				n (%)	Lupane District			
			UM	SM	RR; 95% CI	*p*-Value		UM	SV	RR; 95% CI	*p*-Value
Travel History	No	1056 (87)	937 (86)	119 (98)	1		1056 (87)	808 (85)	248 (99)	1	
	Yes	151 (13)	149 (14)	2 (2)	0.04 (0.01;0.38)	0.004	151 (13)	148 (15)	3 (1)	0.03 (0.01;0.11)	<0.001
Age group	<5 years	77 (6)	61 (6)	16 (13)	1		122 (10)	58 (6)	64 (25)	1	
	5 years +	1130 (94)	1025 (94)	105 (87)	0.18 (0.06;0.62)	0.006	1085 (90)	898 (94)	187 (75)	0.22 (0.13;0.36)	<0.001
					23% (>10%) *					47% (>10%) *	
Sex	Female	670 (55)	688 (62)	2 (2)	1		279 (23)	196 (21)	83 (33)	1	
	Male	537 (45)	418 (38)	119 (98)	126 (24.1;660.4)	<0.001	928 (77)	760 (79)	168 (67)	0.79 (0.54;1.16)	0.227
					75% (>10%) *					26 (>10%) *	
Occupation	Minor	371 (31)	335 (31)	36 (30)	1		425 (35)	320 (33)	105 (42)	1	
	Student	132 (11)	116 (11)	16 (13)	2.23 (0.75;6.75)	0.148	161 (13)	128 (13)	33 (13)	1.13 (0.65;1.95)	0.664
	Unemployed	286 (24)	266 (24)	20 (17)	0.61 (0.20;1.91)	0.400	161 (13)	111 (12)	50 (20)	2.02 (1.22;3.33)	0.007
	Employed	418 (34)	369 (34)	49 (40)	2.23 (0.94;5.38)	0.069	460 (39)	397 (42)	63 (25)	1.00 (0.64;1.56)	0.990
					55% (>10%) *					27% (>10%) *	
Had visitor(s)	No	232 (19)	213 (20)	19 (16)	1		580 (48)	534 (56)	46 (18)	1	
	Yes	975 (81)	873 (80)	102 (84)	0.81 (0.27;2.44)	0.708	627 (52)	422 (44)	205 (82)	6.19 (4.16;9.22)	<0.001
					31% (>10%) *					75% (>10%) *	
Residence	Urban	491 (41)	444 (41)	47 (39)	1		609 (51)	534 (56)	75 (30)	1	
	Rural	716 (59)	642 (59)	74 (61)	0.83 (0.43;1.62)	0.587	598 (49)	422 (44)	176 (70)	1.94 (1.35;2.79)	<0.001
					35% (>10%) *					27% (>10%) *	
Prompt treatment	Within 24 hrs	848 (70)	813 (75)	35 (29)	1		720 (60)	564 (59)	156 (62)	1	
	After 24 h	359 (30)	273 (25)	86 (71)	6.78 (3.34;13.8)	<0.001	487 (40)	392 (41)	95 (38)	1.01 (0.70;1.43)	0.973
Mean ± SD	2.2 ± 2.2				47% (>10%) *		Mean ± SD = 2.6 ± 2.4			32% (>10%) *	
Malaria parasite	Other	26 (2)	26 (3)	0 (0)	--	--	30 (3)	29 (3)	1 (0.4)	--	--
	Malariae	103 (9)	100 (9)	3 (2)	--	--	159 (13)	141 (15)	18 (7)	--	--
	Falciparum	1078 (89)	960 (88)	118 (98)	--	--	1018 (84)	786 (82)	232 (93)	--	--
Malaria contact	Asymptomatic	12 (1)	12 (1)	0 (0)	--	--	2 (0.2)	2 (0.2)	0 (0)	--	--
	Symptomatic	228 (19)	228 (21)	0 (0)	--	--	185 (15)	170 (18)	15 (6)	--	--
	Index	967 (80)	846 (78)	121 (100)	--	--	1020 (85)	784 (82)	236 (94)	--	--
LLIN use	Owned used	369 (30)	357 (33)	12 (10)	1		429 (35)	412 (43)	17 (8)	1	
	Owned unused	347 (29)	318 (29)	29 (24)	24.87 (8.21;75.4)	<0.001	238 (20)	169 (18)	69 (27)	7.83 (4.29;14.3)	<0.001
	None	491 (41)	411 (38)	80 (66)	47.4 (16.4;137.2)	<0.001	540 (45)	375 (39)	165 (65)	12.3 (7.02;21.4)	<0.001
					178% (>10%) *					135% (>10%) *	
Slept outdoors	No	954 (79)	933 (86)	21 (17)	1		598 (49)	488 (51)	110 (44)	1	
	Yes	253 (21)	153 (14)	100 (83)	84.4 (36.1;197.4)	<0.001	609 (51)	468 (49)	141 (56)	1.93 (1.36;2.74)	<0.001
					88% (>10%) *					51% (>10%) *	

1 = Reference: UM = uncomplicated malaria; SV = severe malaria; SD = standard deviation; RR≈ relative risk; “* confounding (>10%)”.

**Table 2 ijerph-21-00877-t002:** Model Evaluation.

Variables	Model I: Overall		Model II: Beitbridge		Model III: Lupane	
	β (*p*-Value)	RR; 95% CI	β (*p*-Value)	RR; 95% CI	β (*p*-Value)	RR; 95% CI
Travel History	−1.70 (0.013)	0.18 (0.05;0.70)	−3.12 (0.004)	0.04 (0.01;0.38)	−3.39 (<0.001)	0.03 (0.01;0.11)
District	−1.00 (<0.001)	0.37 (0.28;0.51)				
Age group	−1.48 (<0.001)	0.23 (0.15;0.35)	−1.70 (0.006)	0.18 (0.06;0.62)	−1.53 (<0.001)	0.22 (0.13;0.36)
Gender	1.02 (<0.001)	2.77 (2.01;3.81)	4.83 (<0.001)	126.1 (24.09;660.4)	−2.24 (0.227)	
Occupation ^1^			0.82 (0.148)		0.12 (0.664)	
Occupation ^2^			−0.49 (0.400)		0.70 (0.007)	2.02 (1.22;3.35)
Occupation ^3^			0.81 (0.069)		0.00 (0.990)	
Had visitor (s)	1.70 (<0.001)	5.45 (3.81;7.80)	−0.21 (0.708)		1.82 (<0.001)	6.19 (4.16;9.22)
Residence	0.49 (0.001)	1.62 (1.21;2.17)	−0.19 (0.587)		0.66 (<0.001)	1.94 (1.35;2.79)
Prompt treatment	0.71 (<0.001)	2.04 (1.53;2.70)	1.92 (<0.001)	6.78 (3.34;13.79)	0.01 (0.973)	
LLIN use ^1^	1.85 (<0.001)	6.34 (3.95;10.2)	3.21 (<0.001)	24.9 (8.206;75.35)	2.06 (<0.001)	7.83 (4.2914.2)
LLIN use ^2^	2.60 (<0.001)	13.5 (8.66;21.0)	3.86 (<0.001)	47.4 (16.38;137.2)	2.51 (<0.001)	12.3 (7.02;21.4)
Slept outdoors	1.63 (<0.001)	5.11 (3.82;6.85)	4.44 (<0.001)	84.4 (36.08;197.4)	0.66 (<0.001)	1.92 (1.36;2.74)
Travel × Sex	−3.35 (<0.001)	0.02				
Omnibus	<0.05			<0.05		<0.05
Hosmer–Lemeshow	0.12			0.17		0.08
PAC (%)	87.8			95.9		85.0
Cox and Snell R-squared	0.26			0.36		0.29
Nagelkerke R-squared	0.45			0.75		0.45
−2 Log likelihood	1341.2			254.2		822.0

## Data Availability

The data in this study are available upon request from the corresponding author.

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
