# Peer review of "Malaria Severity in the Elimination Continuum: A Retrospective Cohort Study between Beitbridge and Lupane Districts in Zimbabwe, 2021–2023"

_ijerph, 2024, doi:10.3390/ijerph21070877_

Round 1

Reviewer 1 Report

Comments and Suggestions for Authors

Introduction

The introduction is comprehensive but requires improvements in clarity and flow. I recommend organizing the information into clear, concise paragraphs, each focusing on a single aspect. Additionally, ensure paragraphs are well-connected by incorporating transitional phrases to guide the reader smoothly through the text. Remove any repetitive information to enhance readability.

Methodology

The methodology section includes the relevant information; however, I have suggested some edits to improve readability and clarity. Please refer to the comments given in the manuscript text.

Results

The results section contains several repetitions, and I have provided specific comments in the text. To improve this section, focus on highlighting the important outcomes and significant findings, as the tables already contain the detailed data. The current presentation of the data is difficult to follow and needs improvement.

Discussion

The discussion section requires major improvements to emphasize the significance of the study. Many outcomes are obvious, and the findings should be specifically addressed and explained with clear reasoning. The current text includes vague statements that weaken the study's strong outcomes. Additionally, some citations used for comparison are inappropriate. Please refer to the comments in the text and rewrite the discussion accordingly.

Author Response

We have added a template that details the responses to Reviewers as a word document

Reviewer 2 Report

Comments and Suggestions for Authors

Thank you for the opportunity to review this manuscript. The authors reported that travel history was unlikely to be associated with severe malaria in two districts in Zimbabwe after analyzing secondary data.

I just have a few questions and comments for consideration.

Revise abstract and omit detailing SPSS

The Introduction needs to be concise and focus more on malaria in Zimbabwe rather than the global aspects of malaria.

Lines 383 -384 Perhaps VFR in the Belgium study are international travelers visiting endemic region from non-endemic countries.

Would it be possible to include the identified areas of malaria endemicity in Figure 1?

The content seems to bleed into subsequent sections. Consider reorganizing the paragraphs to maintain clear section boundaries.

Some content is repetitive. Please correct this throughout the manuscript.

Use one or two tables within the main text, and consider adding the remainder as supplementary materials. Maybe just only use Table 3 in the main text.

Add a reference to the first sentence of the Introduction. Clarify whether it pertains to the resurgence or low transmission of malaria.

How was malaria diagnosed? Please provide details.

Italicize the name of the malaria species on line 55.

Consider moving lines 205-208 to page 16

Consider moving lines 155-158 to the Introduction.

Include operational definitions for terms such as severe malaria, exposed, and unexposed.

Table 1 use either one meanSD or medianIQR, why age group stratified as younger and older than 5 years. Maybe regroup as children and adults. Revise "working" to "employed" 

Table 2. What is meant by other species? and "Travel*Sex"

Please consider adding subheading to the Discussion for reader clarity.

Line 409 what is the meaning of night-time beds (sleeping outdoors vs indoors). Please also provide a reference. 

The authors used binary logistic regression, and would be better to mention specific details such as age group (>5 years), gender.

Is categorizing age groups as <5 and >5 in view of severe malaria?

Please also include the type of occupations in the working groups, as malaria can depend on the type of occupation, for example, forest-related or non-forest-related.

What was the reason for excluding simple regression analysis 

Author Response

We have developed a template and uploaded (detailed response to Reviewer's comments point by point
